# Evaluation of Whole Pigweed Stalk Meal as an Alternative Flour Source for Biscuits

**DOI:** 10.3390/foods14223924

**Published:** 2025-11-17

**Authors:** Zlatin Zlatev, Stanka Baycheva, Toncho Kolev, Svetoslava Terzieva, Neli Grozeva, Milena Tzanova, Dessislava Dimitrova, Teodora Ivanova

**Affiliations:** 1Faculty of Technics and Technologies, Trakia University, 38 Graf Ignatiev Street, 8602 Yambol, Bulgaria; stanka.baycheva@trakia-uni.bg (S.B.); toncho.kolev@trakia-uni.bg (T.K.); 2Faculty of Agriculture, Trakia University, Studentski Grad Str., 6000 Stara Zagora, Bulgaria; svetoslava.terzieva@trakia-uni.bg (S.T.); n.grozeva@trakia-uni.bg (N.G.); milena.tsanova@trakia-uni.bg (M.T.); 3Department of Plant and Fungal Diversity and Resources, Institute of Biodiversity and Ecosystem Research, Bulgarian Academy of Sciences, 1113 Sofia, Bulgaria; dessidim3010@gmail.com (D.D.); teoivan@abv.bg (T.I.)

**Keywords:** amaranth green powder, butter biscuits, functional additives, physico-chemical characteristics, sensory characteristics, nutritional enhancement

## Abstract

In this study, one of the main problems related to the development of new foods and the improvement of existing staple foods is examined. The effect of stalk pigweed flour in relation to the main raw material, wheat flour, at levels of 0%, 5%, 10%, and 15% was evaluated with respect to key characteristics of flour mixtures, dough, and biscuits with this additive. A selection of informative features was made, revealing that out of the 39 studied parameters, covering sensory, physicochemical, geometric, colorimetric, and spectral characteristics, only 19 proved to be informative. Principal component analysis showed that the relationship between amaranth green powder (AGP) concentration and the first two principal components explained up to 99% of the variance. The optimal addition level of 7.17% AGP was identified based on the convergence of biscuit characteristics. pH decreased from 6.55 to 6.28, electrical conductivity increased from 1075 to 3759 µS/cm, and sensory scores for aroma and taste peaked near 7% before declining at higher concentrations. It was demonstrated that the relationship between the amount of amaranth in biscuits and the first two principal components can be described with up to 99% accuracy. It was determined that the optimal amount of amaranth flour in biscuits is +7.17%. The results obtained provide a basis for further research into the rapid automated analysis of biscuits with added pigweed flour, which will contribute to the development of new foods with improved characteristics. It is suggested to carry out more research to study the effect of flours from other amaranth types, enhancing different varieties and cultivated in diverse ecological regions. This work also explores the viability of pigweed as a nutritious and sustainable flour alternative while providing a multivariate approach in view of newly developed bakery formulations.

## 1. Introduction

The quest for new sources of nutritional and bioactive substances has increased in recent decades [1]. Pseudocereals, like amaranths, are a good alternative to traditional cereal crops for their potential in the development of gluten-free foods and/or those with a low glycemic index [1,2]. They are a good complement to other grain crops in arid and semi-arid areas that do not thrive well [3,4]. Amaranths are highly adaptable to a wide range of ecological conditions due to a set of versatile physiological traits, genetic diversity, and phenotypic plasticity. Still, in many cases they are considered underutilized [5]. In countries where amaranths are grown for their grain, the stems are usually considered as agricultural waste; however, they are rich in fibers, microelements, and biologically active substances [6]. In the face of global climate changes, finding new and diverse ways to tap these resources would contribute to green transition efforts [7].

The genus *Amaranthus* L. (Amaranthaceae) comprises 70 widely distributed annual and perennial species [8]. Amaranth seeds and green biomass are a part of the traditional cuisines of Northern America, the Indian subcontinent, Africa, and some European countries [9,10,11]. *Amaranthus hypochondriacus* (L.) B.L.Rob., *A. cruentus* L., and *A. caudatus* L. are cultivated for their seeds (known as grain amaranths) [10,12,13,14,15], while *A. hybridus* L., *A. tricolor* L., *A. viridis* L., *A. blitum* L., *A. dubius* Mart., *A. graecizans* L., and *A. retroflexus* L. are known as leafy vegetables [11,16,17]. *Amaranthus caudatus* L., *A. hybridus* L., and *A. tricolor* L. are also decorative and *A. retroflexus* L. and *A. albus* L. are applied in some traditional medicine systems [18,19,20].

The inclusion of fruit and vegetables in bakery goods, especially raw ingredients that are potentially destined as agricultural waste, is recognized as a valuable source of functional dietary fibers and phenolic compounds [21]. Blended fresh and dry leafy vegetables were successfully used as fortification agents in various bakery products, improving mineral content and fatty acid and amino acid profiles, as well as contributing as antioxidant agents [22]. Amaranths were reported to contain valuable nutritional and mineral compounds, some essential amino acids, carotenoids, antioxidants (e.g., hydroxycinnamic acid), etc. The works of [22,23,24] presented a wide survey of nutritional and therapeutic features of *Amaranthus* spp. and their potential as ingredients in various food products, among which are baked foods and biscuits. Amaranth seeds and related products (flours, coarse blends, etc.) are well-studied and common ingredients of various gluten-free foods [6,25]. On the other hand, the potential of the remaining parts of the plant—roots, leaves, and herbage—is less studied as a functional additive in the food industry. Amaranths are known as traditional leafy vegetables worldwide, but they are more seldom used as ingredients for bakery goods [26,27]. As seeds were the primary use of these crops in their area of origin, the Americas, some authors even consider the secondary use of amaranths as vegetables to be related to their accidental distribution as weeds and the ignorance of local communities towards usage of the seeds [28].

The whole above-ground part of the amaranth plant could be used as an additive to basic foods like bread, cakes, biscuits, porridge, and pasta as a way for efficient utilization of green biomass from agriculture [29,30]. Amaranth greens, usually young leaves, are cooked like spinach in stews or soups and added to salads or traditional pastries [11,31,32]. Pastries with amaranth greens in Bulgaria were part of Lenten fare, using young plants as filling or directly as part of the filo dough [11,31]. Still, consumption of wild amaranth greens in Europe was mostly in remote areas with poor soils and as emergency/wartime food when organoleptic qualities were least prioritized [31].

As species availability and local preferences towards the developmental stage of the plants vary, raw amaranth collected from the wild and/or available on the market could have very diverse nutritional and gustatory properties. Biscuits made with red morphs of *A. tricolor* leaves were reported to have significantly more protein and fibers than the control [30]. Compared to spinach pasta with added *A. caudatus,* leaf powder had insignificantly higher content of fats and protein [33]. Same authors report on similar sensory ratings for both spinach and amaranth, with only half of the panelists recognizing amaranth as an ingredient. Green leaf powder of *A. cruentus* added to corn snacks resulted in elevated protein and fiber content, and hence harder products, which was negatively perceived by sensory panelists, especially in cases of higher percentages of amaranth leaf powder (3%) [34]. Hiscock et al. [35] reported that even boiled leaves of genotypes in the same *Amaranthus* species were ranked with both similar and contradictory sensory properties [35]. Same authors outline that panelists most frequently describe the samples as “leafy”, followed by “spinach”-like, “soft”, and “tasteless”, which could be improved with the addition of other vegetables. When incorporated in noodles, amaranth leaf powder (from botanical unspecified market origin) was found to significantly increase the fat content whilst reducing the amounts of total glycemic carbohydrates [36], with the taste described as “leafy” also [37]. Salty muffins with up to 6% amaranth leaf powder had no statistically significant difference regarding the sensory parameters to the control group [38]. Conversely, the addition of amaranth leaf powder (7% of the dough) was shown to significantly increase total phenolic content but received the lowest sensory rates compared to the control and other vegetable breads [39]. The aim of the current study is to assess the qualities and organoleptic characteristics of salty butter biscuits with different amounts of amaranth green powder (AGP) prepared from different wild *Amaranthus* species growing in Bulgaria and to establish the optimal quantity to be included in biscuits in order to improve their nutritional value whilst preserving acceptable sensorial and physical characteristics.

Unlike most studies that focus on amaranth seeds or cultivated varieties, this research utilizes the whole above-ground biomass of wild *Amaranthus* species, typically considered agricultural waste, and applies a multifactorial evaluation approach combining physico-chemical, geometric, colorimetric, spectral, and sensory analyses to determine optimal flour substitution levels in biscuits.

## 2. Materials and Methods

### 2.1. Plant Material

The AGP contains equal parts from five species: *Amaranthus albus*, *A. blitum*, *A. deflexus*, *A. hybridus*, and *A. retroflexus*. These are widely distributed in Bulgaria and plants were collected from 29 wild populations from six floristic regions: Central Stara Planina Mt., Thracian lowlands, West Frontier Mts., Strandzha Mt., Tundzha Hilly plain, and West Rhodopi Mt. The collected plant material (whole herbage) was sorted, inspected for damages, washed under running water, and then dried at room temperature with natural ventilation under shade conditions. After drying, the herbage was turned into powder with a high-speed multifunctional blender model BN1200AL (Gorenje PLC, Velenje, Slovenia). The powder samples were stored in paper bags in dark and aerated place at 16–22 °C.

### 2.2. Biscuit Formulation and Methodology of Preparation

The following ingredients were used for the preparation of the biscuits: type 500 all-purpose white flour (Topaz Mel Ltd., Sofia, Bulgaria), iodized salt, 82% cow butter (Milky group Ltd., Haskovo, Bulgaria), and hen egg yolks obtained according to Ordinance 1/2008 on the requirements for the marketing of table eggs. For the experiment, only the quantities of the flour and the AGP were changed; all other ingredients were constant (Table 1).

The stages of preparation for the salty butter biscuits with added AGP are given in Figure 1. Some of them are conducted at room temperature; refrigeration was used so as to avoid melting the butter. The total technological time to prepare the butter biscuits is 2.5–3 h.

### 2.3. Testing of Physico-Chemical Parameters

#### 2.3.1. Measurement of pH, Electrical Conductivity (EC), Total Dissolved Solids (TDS), and Oxidation-Reduction Potential (ORP)

The protocol described in the standard method, AACC 02-52.01 [40], was applied. Briefly, a 5 g sample was added to 50 mL distilled water at 70 °C. The mixture was homogenized by stirring and left to cool down to room temperature. Three independent measurements of every parameter were conducted and the mean and standard deviation were calculated.

The quantities of the samples were measured with Pocket Scale MH-200 (Zhezhong Weighing Apparatus Factory, Yongkang City, China). pH; EC, µS/cm; TDS, ppm was measured with a multifunctional device PH-TDS-EC-TEMP Meter (Nanjing Tsung Water Technology Company Ltd., Nanjing, China).

The protocol described in AACC Method 02-52.01 [40] was used for sample preparation. The oxidation–reduction potential (ORP, mV) of the aqueous extract was then measured potentiometrically using a platinum electrode with an Ag/AgCl reference electrode Model ORP-2069 (Shanghai Longway Optical Instruments Co., Ltd., Shanghai, China).

#### 2.3.2. Temperature Measurement (T)

T, °C was measured with a digital thermometer V&A VA6502 (Shanghai Yihua V&A Instrument Co., Shanghai, China).

#### 2.3.3. Determination of Loss After Baking

After baking, the biscuits were cooled down to room temperature and after that they were weighed on a technical scale Boeco BBL-64 (Boeckel GmbH, Hamburg, Germany), with a maximum capacity of 300 g and a resolution of 0.1 g. The losses were calculated with the following formula: (1)$$TL=\frac{a-b}{a}\times 100,\;\%$$
where *TL*, % are thermal losses; *a* is the weight before baking, g; *b*—the weight after baking, g.

#### 2.3.4. Determination of the Spread Factor After Baking

To determine the spread factor (*SF*), the diameter (*D*, mm) and height (*h*, mm) were measured after baking. The diameter and height were measured in three different positions and the mean values were calculated. A digital caliper model SEB-DC-023 (Shanghai Shangerbo import & export Co., Ltd., Shanghai, China) was used, with a maximal length measurement of 150 mm and a resolution of 0.05 mm (Shanghai Shangerbo import & export Co., Ltd., Shanghai, China). Spread factor was determined with the following formula:(2)$$SF=\frac{D}{h}$$
where *D* is the diameter of the biscuit after baking, mm; *h*—height of the biscuit after baking, mm. The obtained value has no dimension.

### 2.4. Organoleptic Analysis

The organoleptic analysis of the biscuits was performed according to the Bulgarian State Standard EN ISO 13299:2016 (Sensorial analysis. Methodology. General manual for establishing sensorial profile) [41].

The sensory panel consisted of nine evaluators (five academic staff and four graduate students) from the Food Technologies Department, selected based on their prior training in sensory evaluation and familiarity with biscuit formulations. The tasting was conducted in a controlled environment using individual booths under standardized daylight-balanced lighting (D65 equivalent) to minimize visual bias. Samples were coded with randomized three-digit numbers and presented in a randomized order to avoid positional effects. Panelists were instructed to cleanse their palates with water between samples to ensure consistent evaluation across all biscuit variants.

The requirements for the code of ethics of the university community were met (http://uni-sz.bg/truni5, accessed on 25 June 2022, in Bulgarian).

The butter biscuits examined in this study are confectionery products compliant with the Bulgarian national standard BNS 441:1987 (Biscuits. General requirements) [42], which defines biscuits as products suitable for human consumption. The pigweed (*Amaranthus* spp.) used as an ingredient is also recognized as food-grade according to Bulgarian Regulation No. 5 of 3 September 2018 on organic production, labeling, and control of plant products. Sampling and organoleptic analyses were conducted in accordance with Bulgarian Regulation No. 2 of 27 March 2024 and Regulation No. 12 of 10 November 2021, which govern food sampling and laboratory testing and do not require ethical approval when no human or psychological factors are involved. Organoleptic evaluation followed the international standard BDS EN ISO 13299:2016, which specifies sensory analysis methods for food and does not involve the collection of personal data or medical information from participants.

A 5-point Likert scale was used to evaluate the biscuits (1—completely does not correspond to the indicator; 5—completely corresponds to the indicator). Finally, an overall average score of the organoleptic characteristics of the biscuits was calculated.

### 2.5. Color Analysis

The color digital images were acquired with the video sensor of a mobile phone LG L70 (LG Electronics, Inc., Seoul, Republic of Korea). The sensor is VB6955CM (STMicroelectronics International N.V., Geneva, Switzerland), resolution 2600 × 1952 pix and 1.4 × 1.4 μm pixel.

Color digital images were obtained in RGB color model, converted to Lab color model according to CIE Lab 1976. Color component conversion functions at observer 2o and illumination D65 were used.

The color difference Δ*E* of the biscuits with added amaranth green powder was determined. It varied in the range 0–100; biscuits with color close to 0 were closer to those of the control sample, while those that were closer to 100 differed from the control sample. Color difference below 20 was difficult to distinguish with the naked eye.(3)$$∆E=\sqrt{{\left(L_{c}-L_{a}\right)}^{2}+{\left(a_{c}-a_{a}\right)}^{2}+{\left(b_{c}-b_{a}\right)}^{2}}$$
*L_c_*, *a_c_*, *b_c_* were color components of the control sample; *L_a_*, *a_a_*, *b_a_*—components of the sample with the additive.

The following color indices were calculated: *C_1_* indicates the brightness of the brown color. Brown nuances normally have moderate-to-low *L* value, slightly positive *a* value and moderately high *b* value; *C*_2_ indicates the darkness of the brown color with special reference to lower brightness; *C*_3_ indicates the yellow-brown nuances with specific reference on the yellow component but preserving the brown nuance as well; *C*_4_ indicates the density of the brown color and refers to very dark brown nuances.

The color indices have the following formulae:(4)$$C_{1}=0.5\times L+\left(50-a\right)+\left(b-5\right)$$(5)$$C_{2}=\left(100-L\right)+2\times b-0.5\times a$$(6)$$C_{3}=b+\left(100-L\right)-a$$(7)$$C_{4}=2\times \left(100-L\right)+b-a$$

Spectral characteristics were obtained. The conversion of the values from XYZ and LMS models to reflectance spectra in the VIS region, in the 390–730 nm range, was performed using mathematical formulas presented by Vilaseca et al. [43]. These formulas were for observer 2^o^ and illumination D65.

The following spectral indices were calculated: *S*_1_ emphasizes the balance between red and green, which can be useful in distinguishing between reddish-brown and greenish-yellow colors; *S*_2_ measures the steepness of the spectral curve between two wavelengths. It shows the dominance of certain color transitions useful for analyzing color changes in the spectrum; *S*_3_ measures the depth of a reflection feature useful for identifying colors with strong absorption in specific spectral regions.

The spectral indices were determined at two wavelengths, 530 and 630 nm, applying the following formulas:(8)$$S_{1}=\frac{R_{630}}{R_{530}}$$(9)$$S_{2}=\frac{R_{530}-R_{630}}{20}$$(10)$$S_{3}=\frac{R_{530}-R_{630}}{R_{530}+R_{630}}$$

### 2.6. Statistical Analysis

The data were statistically processed with Matlab 2017b (The Mathworks Inc., Natick, MA, USA). Preparation and visualization of data were performed with MS Office 2016 (Microsoft Corp., Washington, DC, USA).

All measurements were made in three independent repeats and the means were used in the tables and graphs. The statistical functions of standard deviation and maximum allowable error were used. All data were processed at an accepted significance level of α < 0.05.

One-way ANOVA was used to analyze the data. A post hoc LSD test was also performed to determine the degree of significance of differences between average values after statistically significant differences (*p* < 0.05) were found. The non-parametric Kruskal–Wallis test was used when there was no normal distribution.

#### 2.6.1. Formation of Feature Vectors

Feature vectors that describe the flour mixtures (F1–F4), the dough (F5–F16), and the biscuits (F17–F39) were established. The features and their meaning are given in Table 2.

#### 2.6.2. Selection of the Informative Features

For the selection of the informative features, the RReliefF method was used. The ReliefF method registers relations between the features and is noise resilient. The features with a weight coefficient above 0.6 are considered as informative [44]. A vector was defined from the selected features. In this context, informative variables are those features that provide the most relevant and discriminative information for distinguishing among the analyzed samples, as identified by their high ReliefF weights.

#### 2.6.3. Principal Component Analysis (PCA)

The data from the feature vector were reduced using the PCA method [45]. It is a statistical method to reduce the dimensionality of data. A large set of data can be processed and transformed into a smaller set of variables, the so-called principal components, each of which is a linear combination of the original variables arranged according to the depression size derived from the data.

Possible combinations between % additive and different informative features are given in Table 3. Such analysis was performed for all cases of added green amaranth powder (from 0 to 15%), and only the selected informative features were used.

A regression model more frequently used for analysis of food products was applied [46]. It describes the relation between independent and dependent variables according to the following formula:(11)$$z=b_{0}+b_{1}x+b_{2}y+b_{3}x^{2}+b_{4}xy+b_{5}y^{2}$$
*z* is the dependent variable, while *x* and *y* are the independent variables; *b* refers to the model coefficients.

For the assessment of the model, the determination coefficients (R^2^), model coefficients, their standard error (SE), *p*-values, and Fisher criterion (F) were used. Analysis of the residues was performed.

To define the correct quantity of the amaranth green powder additive a linear programming algorithm was applied using the function “linprog” in Matlab 2017b (The Mathworks Inc., Natick, MA, USA) Software System. Linear programming involved finding the solution for a vector *x* so that the linear function *f^T^x*, with linear constraints(12)$$f^{T}x\;$$
fulfills one of the following conditions:(13)$$Ax\leq b\;\;\;\;\;\;\;A_{eq}x=b_{eq}\;\;\;\;\;\;\;\;l\leq x\leq u$$
where *f*, *x*, *b*, *b_eq_*, *l*, and *u* are vectors, and *A* and *A_eq_* are matrices.

An “*Interior-point-legacy*” function was used. The function arrives at an appropriate solution by traversing the inside of the data area.

## 3. Results

### 3.1. Composition of the Amaranth Green Powder

The main characteristics of the green powder of the used amaranth species are presented in Table 4.

### 3.2. Analysis of the Amaranth Green Powder and the Mixtures of Wheat Flour and AGP

The results showed that the AGP mixture had a slightly acidic pH, high TDS and EC values, and low ORP values (Table 5). The data suggest a high mineral content of AGP, which may have an impact on both oxidative stability and nutritional value when incorporated into butter biscuits.

Statistically significant differences were registered between the different flour–AGP mixtures (from 0% to 15%). (Table 5) The increase in the AGP content caused a slight decrease in the pH value (from 6.55 to 6.28); hence, the acidity of the mixture slightly raised.

With an increase in the amount of the additive, the concentration of dissolved ions and minerals in the flour also rises, as evidenced by significant increases in both TDS and EC. TDS increases from 627 ppm at 0% additive to 2523 ppm at 15%, and EC rises from 1075 µS/cm to 3759 µS/cm over the same range, reflecting an improvement in ionic strength and mineral content. Furthermore, ORP also increases from 65 mV to 78 mV, indicating that higher additive concentrations lead to a shift toward a more oxidative environment.

The results indicate that the additive strongly alters the physicochemical properties of the flour, making it more acidic, more mineral-rich, and more oxidatively active, which could, in turn, affect its behavior during baking as well as its nutritional value.

### 3.3. Analysis of Dough

The physicochemical properties of biscuit dough containing different additive levels (0%, 5%, 10%, and 15%) are summarized in Table 6. The pH remained largely consistent across all formulations (6.24–6.35), indicating near-neutral conditions with slight acidity. From Table 6 it is evident that both TDS and EC increased with rising additive concentrations, reflecting higher levels of dissolved minerals and enhanced ionic strength. Specifically, TDS increased from 1834.5 ppm at 0% to 2439 ppm at 15%, while EC rose from 2962.5 µS/cm to 3932.5 µS/cm, reflecting enhanced ionic strength. ORP values remained relatively stable around 65 mV, suggesting that the additive has minimal effect on the dough’s redox balance. Overall, these results indicate that the additive enhances mineral content and ionic activity without significantly altering pH or redox stability, preserving the chemical integrity of the dough.

Figure 2 illustrates the overall changes in the surface characteristics of butter biscuit dough with increasing levels of amaranth flour. As the amount of plant flour increases, both the color characteristics and the surface structure of the product are noticeably altered.

Figure 3 presents the Lab and spectral characteristics of biscuit dough containing amaranth. Lab values and spectral data generally change according to the color properties of the additive as its proportion in the dough increases. With higher additive levels, the “L” value decreases, indicating that the dough becomes darker. The “a” and “b” values adjust according to the hue of the additive: “a” decreases because the additive lacks a reddish tint, while “b” increases due to the additive’s greenish-yellow hue. The resulting spectral data reflect the loss of lightness, accompanied by wavelength shifts corresponding to the reflectance characteristics.

Figure 4 illustrates the color difference (ΔE) of biscuit dough relative to the control (0% additive) across varying additive levels. Increasing the additive concentration resulted in progressively larger ΔE values, indicating color changes that are readily perceptible to the naked eye.

The color index values of biscuit dough with varying percentages of additive are presented in Table 7. Increasing the additive content resulted in significant changes, highlighting its strong impact on the dough’s appearance. With higher additive concentrations, the C1 value decreased, indicating a reduction in the intensity of this color index, while C3 and C4 values increased, reflecting enhanced color characteristics associated with these indices. C2 initially increased, reaching a maximum at 10% additive, before slightly declining, suggesting a more complex effect on this particular index. Overall, the additive substantially altered the color profile of the dough, enhancing or, in some cases, diminishing specific colors at different concentrations.

The values of the spectral indices of biscuit dough with different percentages of additive are presented in Table 8. The data show significant variations in the indices with changes in additive levels. Index S1 remains constant across all additive concentrations, indicating that this spectral characteristic is not significantly affected. Index S2 decreases sharply from 0.87 at 0% additive to 0.24 at 15% additive, indicating a substantial change in the spectral properties of the dough due to the additive. Index S3 remains practically unchanged across different additive levels. In conclusion, the additive has a strong impact on the spectral index S2, while the indices S1 and S3 remain stable and are not significantly affected.

### 3.4. Biscuit Analysis

The main physicochemical characteristics of the biscuits show that increasing the amount of amaranth flour results in a slight decrease in pH. TDS and EC are strongly increased, while ORP remains approximately constant across all levels of the additive (Table 9).

Table 10 presents data on the changes in the geometric characteristics of the biscuits and their heat losses during baking. Biscuit diameter remains relatively constant across all additive levels, while height increases slightly, peaking at 10% additive before slightly decreasing at 15%. Variations in the spread factor are minor, with no clear trend observed. Heat losses rise with increasing additive content, reaching a maximum at 10%, and stabilize at 17% for 5% and 15% additive levels.

The results of the sensory evaluation of biscuits with varying levels of amaranth flour are presented in Table 11.

Increasing the additive by 5% generally leads to a decline in sensory attributes, including appearance, texture, aroma, taste, and chewiness. At 10% additive, however, most of these attributes improve, reaching values comparable to the control (0% additive). Further increasing the additive to 15% causes a slight decrease in sensory scores. These findings suggest that incorporating up to 10% amaranth flour achieves an optimal balance between enhanced sensory qualities and overall acceptability, while higher levels may negatively affect the sensory profile of the biscuits (Figure 5).

Color digital images of the resulting biscuits are shown in Figure 6. Increasing the amount of amaranth flour in the biscuits visibly alters both the color characteristics and the surface structure of the product.

Figure 7 illustrates the Lab and spectral characteristics of biscuits containing amaranth. Increasing the amount of amaranth flour leads to a decrease in the “L” value, indicating that the biscuits darken as the additive level rises. This color component decreases from 56.81 at 0% additive to 36.90 at 15%. The “a” component (red/green) has negative values, indicating that the color lies within the green spectrum. With increasing additive levels, “a” becomes more negative, changing from −4.74 at 0% to −6.33 at 15%, showing a slight intensification of the green hue. The “b” component (yellow/blue) increases with the amount of additive, reaching a maximum of 39.15 at 10%, and then slightly decreases to 35.37 at 15%. This suggests that the yellowing of the biscuits increases up to a certain point and then slightly decreases at the highest additive level.

The VIS reflection spectra reveal a general decrease in reflectance with increasing levels of the additive, indicating progressive darkening of the biscuits due to higher light absorption. A slight increase in reflectance is observed in the green wavelength range, which becomes more pronounced at higher additive levels. Reflectance in the yellow wavelength range rises up to 10% additive, then decreases slightly at 15%, suggesting a reduction in yellow hue intensity. Overall, the spectra indicate a trend toward darker and more muted colors with increasing additive content.

Figure 8 shows the color difference (ΔE) relative to the control biscuit sample, depending on the amount of amaranth flour. The ΔE results indicate significant color changes with increasing additive levels, reaching a maximum of ΔE = 63.27 at 5% additive. As the additive percentage increases to 10% and 15%, the color difference slightly decreases to ΔE = 55.88 and 51.51, respectively, suggesting that the initial addition of the additive causes a more pronounced color change, while higher concentrations produce a less marked effect. This trend indicates that the most significant variation in color occurs at lower additive levels, with the rate of change decreasing thereafter, likely due to saturation effects or interactions among the ingredients in the biscuit formulation.

Table 12 shows the color index values of biscuits containing varying amounts of amaranth flour. The C1 index increases at 5% additive but decreases with higher concentrations, indicating a gradual loss of brightness. The C2 and C3 indices rise progressively at 10% additive, reflecting enhanced color intensity and saturation, and continue to increase up to 15%, indicating further color deepening. These results suggest that moderate additive levels, around 10%, provide optimal color characteristics, while higher concentrations lead to more complex or less desirable color changes.

Table 13 presents the spectral index values of biscuits containing varying percentages of amaranth flour. The indices S1 and S2 exhibit clearly differentiated trends with increasing additive concentration. S1 shows a slight increase up to 10% additive, followed by a decrease at 15%, indicating a positive effect that levels off at higher concentrations. In contrast, S2 decreases markedly as the additive concentration rises, suggesting a reduction in the measured property (e.g., color or texture) with higher amaranth flour content. S3 shows minimal variation, with a slight overall decrease and a small increase at 15%, reflecting a negligible effect of the additive.

### 3.5. Determination of the Optimal Amount of Pigweed Flour in Biscuits

Figure 9 presents the results of informative feature selection using the RReliefF method. Features with weight coefficients above 0.6 indicate that TDS, EC, and ORP are the most influential variables in this model, showing significant changes across different additive levels, with weights of 0.97 and 1.00. Other important features include C1, C3, C4, S2, and ΔE, all exhibiting high coefficients ranging from 0.85 to 0.92. In contrast, features such as pH and ORP have relatively low coefficients in some cases, suggesting that their changes may be context-dependent. Furthermore, features with high weight coefficients but relatively low mean values, such as C2 and C4, are variably affected under different conditions.

The following vector of informative features is formed:

(14)
FV = [F2 F3 F4 F6 F7 F9 F11 F12 F14 F16 F19 F21 F22 F23 F24 F26 F27 F28 F34]


An analysis of principal components was conducted to investigate the relationship between % additive and the selected biscuit features. Prior to analysis, all data were normalized to the [0, 1] range. Table 14 presents the normalized values of % additive relative to the selected features.

TDS shows the highest normalized values for F2 and F3, reaching 0.75 and 0.71 at 15% additive, respectively, indicating that both features gain importance as the additive level increases. In contrast, ORP exhibits minimal variation, remaining low across all additive levels. Other features, such as C1 (F12) and C2 (F14), display more variable behavior: C1 peaks at 0% additive with a value of 0.71 and declines slightly at higher levels, while C2 shows low variability. Changes in ΔE (F16) and C (F34) are particularly pronounced: ΔE rises sharply to 0.98 at 15%, whereas C (F34) jumps from 0.00 to 0.73 at 10% additive before decreasing. The observed variability among features highlights that each characteristic differs considerably in its relevance and effectiveness relative to additive level, underscoring the need for targeted analysis to optimize additive use.

The results of the PCA are presented in Figure 10. Increasing the proportion of amaranth flour in the biscuits was found to significantly influence several key characteristics, including the color difference of both the dough and biscuits (ΔE—F16, F28) and biscuit consistency (C—F34). Furthermore, as the level of supplementation increased from 0% to 15%, a positive shift in *PC*_1_ was observed, indicating a pronounced effect of the additive on the overall product characteristics. Among these, the TDS of the dough (F6) exhibited a moderately positive response. Similarly, the EC values of the flour blends, dough, and biscuits (F3, F7, and F19) were moderately positively affected by the additive. Conversely, negative effects were observed in the ORP of the flour blends (F4) as well as in certain color attributes of the dough and biscuits, particularly lightness and browning (C1—F9 and F21). Overall, *PC*_1_ accounted for the largest proportion of the additive’s influence on biscuit properties, explaining 69.96% of the total variance. This demonstrates the relevance of these parameters in describing the impact of the additive on biscuit quality.

These findings provide valuable insights for formulation strategies aimed at optimizing biscuit quality in accordance with the desired outcomes associated with the incorporation of amaranth flour.

A regression model of the form *%A* = *f*(*PC*_1_, *PC*_2_) was obtained. Non-informative coefficients with *p* > α were removed. The model equation is as follows:(15)$$\%A=-0.06+8.77{PC}_{1}-28.15{PC}_{2}+15.71{{PC}_{1}}^{2}-29.92{{PC}_{2}}^{2}+49.1{PC}_{1}{PC}_{2}$$

Table 15 presents the evaluation of the obtained model. The coefficient of determination was R^2^ = 0.99. According to the Fisher criterion, F(5, 6) = 611.2 >> F_critical = 4.39, with a significance level of *p* << 0.05. The standard error was SE = 0.35.

The regression model predicting *%A* from the two principal components, *PC*_1_ and *PC*_2_, and their interactions exhibits excellent predictive strength, with R^2^ = 0.99. This indicates that the model explains 99% of the variance in *%A*. All predictors and their interactions were significant at *p* < 0.001. *PC*_1_ has a positive effect on *%A*, whereas *PC*_2_ has a negative effect. The interaction terms *PC*_1_^2^ and *PC*_2_^2^ indicate that the effects of *PC*_1_ and *PC*_2_ increase or decrease with their respective magnitudes.

Residual analysis showed that the residuals closely align with a straight line on the normal probability plot and are approximately normally distributed. These results provide strong evidence that the regression model describes the experimental data with sufficient accuracy.

The appropriate amount of amaranth flour in the biscuits was determined, and the result is visualized in Figure 11. The figure presents model form. The star is the point from the model with the appropriate amount of amaranth additive in wheat flour. The presented value is the appropriate amount of additive. The positive value of *PC*_1_ indicates that the optimal 7.17% additive preserves certain properties more closely to the lower percentages (0%, 5%, 10%). This renders the supplemented biscuits more similar to the characteristics of the control sample. In this context, these properties are more closely associated with weight, diameter, spread ratio, and certain sensory attributes such as taste and aroma of the control sample. The second principal component, *PC*_2_, is close to zero, indicating minimal or no deviation at the 7.17% additive level. This suggests that critical biscuit characteristics are maintained without extreme changes, resulting in a product that is neither over- nor under-modified while retaining the beneficial properties of the additive.

The results indicate that the addition of varying amounts of amaranth flour strongly influences the color and structure of the biscuits. As described in the analysis, increasing the level of supplementation leads to significant changes in certain biscuit characteristics, such as color difference and the organoleptic attribute of consistency. Positive changes in *PC*_1_ suggest that the additive exerts an influence on the properties represented by this principal component, with a moderate positive effect on TDS and EC, and an impact on ORP and color attributes. These findings are consistent with previous reports on the role of amaranth flour in modulating the physical and sensory properties of biscuits [47,48].

Polyunsaturated fatty acids (PUFAs) have demonstrable effects on the texture and aroma of baked goods, and comparative studies support these claims.

The incorporation of PUFA-rich ingredients into baked goods has been shown to influence both texture and sensory perception. For instance, Martínez et al. [49] demonstrated that reformulating puff pastry with oils derived from chia and poppy seeds, both rich in PUFAs, resulted in a softer crumb structure and enhanced aroma, attributed to the oxidative breakdown of unsaturated fatty acids during baking. Similarly, Dominguez et al. [50] reported that structuring PUFA-rich oils with monoglycerides improved lipid dispersion and contributed to a more cohesive, less brittle texture in semisolid bakery matrices. These findings align with the current study’s observation that amaranth green powder, rich in PUFAs, subtly modifies the biscuit texture and aroma, suggesting that the lipid profile of the additive plays a functional role in shaping sensory outcomes.

Comparative sensory analyses further support the impact of PUFA-rich additives on aroma and mouthfeel. Pimdit et al. [51] found that puff pastries formulated with salatrim, a fat replacer containing PUFAs, exhibited higher moisture retention and softer texture, though some sensory attributes were less pronounced compared to conventional fat formulations. Moreover, the American Society of Baking notes that PUFAs, despite their susceptibility to oxidation, are valued in commercial baking for their ability to enhance flavor complexity and contribute to a lighter mouthfeel [52]. These parallels reinforce the current study’s findings that moderate inclusion of amaranth green powder (around 7%) optimizes both nutritional and sensory characteristics, aligning with broader trends in functional bakery product development.

The optimal additive level, determined as 7.17% via PCA, represents the point at which the desired properties closely resemble those of the control sample, maintaining close alignment with characteristics such as weight, diameter, and sensory attributes related to taste and aroma. These results complement observations by Karki et al. [53] and Raihan et al. [54], who reported that lower levels of amaranth flour better preserve sensory qualities while providing nutritional benefits [53,54]. Overall, the findings of this study suggest that 7.17% is the optimal level for balancing the nutritional advantages of amaranth flour with the preservation of the sensory quality of the biscuits.

## 4. Conclusions

This study presents the evaluation of the effects of incorporating varying pigweed stalk (*Amaranthus* spp.) flour on the characteristics of butter biscuits. Several analyses were conducted, and it was found that the addition of 7.17% pigweed flour improved both the physicochemical and sensory properties of the biscuits. Determining the optimal level of pigweed flour provides guidance for product formulation and can enhance overall biscuit quality. The results of this study complement the existing literature on pigweed flour-enriched biscuits by providing data on color and spectral indices, allowing a more detailed and in-depth assessment of the product, as these indices are associated with changes in composition and biscuit properties.

As determined by the simultaneous consideration of physico-chemical parameters and sensory characteristics, 7.17% AGP could be considered the optimal addition. With the increase in AGP levels from 0% to 15%, the pH of the flour mixture reduced slowly (from 6.55 ± 0.06 to 6.28 ± 0.05), which demonstrated that acidification for the flour mixture was very mild. Meanwhile, TDS and EC increased from 627 ± 97 to 2523 ± 243 ppm and from 1075 ± 108 to 3759 ± 421 µS/cm, which indicates more mineral burden with an increase in the ions of the affected areas. The ORP also significantly increased (from 65 ± 19 mV to 78 ± 18 mV), reflecting a shift toward an oxidative environment. These changes may enhance nutritive values, but they might influence dough rheology and final product quality. Sensory also corroborated the 7.17% limitation, as biscuits with AGP over 10% started to diminish on panelists’ scores for aroma, taste, and chewiness, which may be due to increased vegetable notes and firmer texture. Thus 7.17% becomes an equilibrium where nutrients are enriched and without significant deterioration in consumers’ acceptability and essential technological characteristics.

Increasing amounts of pigweed flour could cause biscuits to maintain their diameter, rise in height, and show a higher spread ratio, which was also noted for heat losses. Also, 39 properties of the flour, dough, and biscuits (geometric, physicochemical, color, and sensory) were determined; 19 were selected as the most informative. Geometric parameters, electrical conductivity, firmness, taste, color, and spectral indices were the most pertinent descriptors for characterization of the effect of pigweed flour incorporated in this study. The associations of these 19 features with additive level were detailed by principal component analysis, accounting for up to 99% of the variance. Pigweed flour is appropriate to produce some types of butter biscuits, given its distinct chemical composition and high dietary fiber level.

These results can be used to correct the feeding level of stalk pigweed flour for better product nutrition and functional food development. Considering the diversity of species in the genus amaranthus, additional studies are necessary to evaluate the effect of amaranth-derived flour from other species.

These studies would contribute to further knowledge regarding the nutritional and technological characteristics of the species, cultivated or in natura, as well as its potential use as a functional ingredient by the food industry.

The results from this study demonstrate the industrial potential of the findings in relation to food application. The proved influence of the additive reiterates the value of amaranth as a health food and a rich agri-food with immunity power that enhances the nutritional and functional potentiality in bakery goods. This promotes the environmentally friendly use of agricultural resources and represents a practicable strategy for the development of new products in industrial food production.

## Figures and Tables

**Figure 1 foods-14-03924-f001:**
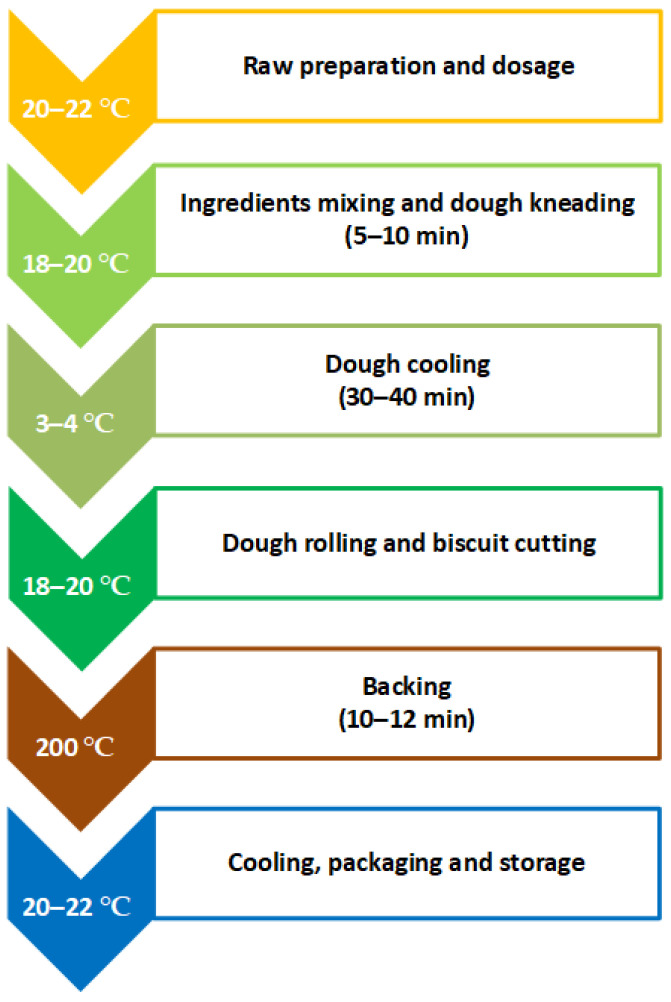
Technology for preparing salty butter biscuits with added AGP.

**Figure 2 foods-14-03924-f002:**

Color images of biscuit doughs with additive. (**a**) 0% additive; (**b**) 5% additive; (**c**) 10% additive; (**d**) 15% additive.

**Figure 3 foods-14-03924-f003:**
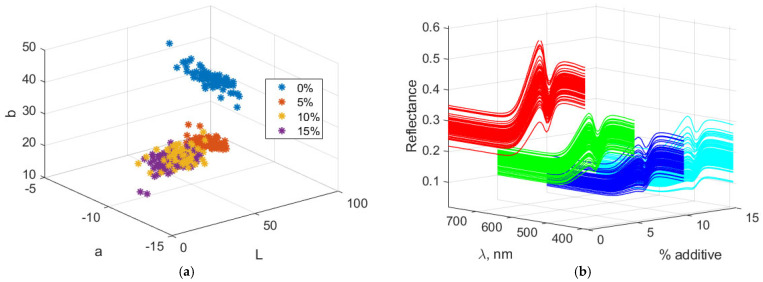
Color and spectral characteristics of biscuit dough. (**a**) Lab color characteristics; (**b**) spectral characteristics.

**Figure 4 foods-14-03924-f004:**
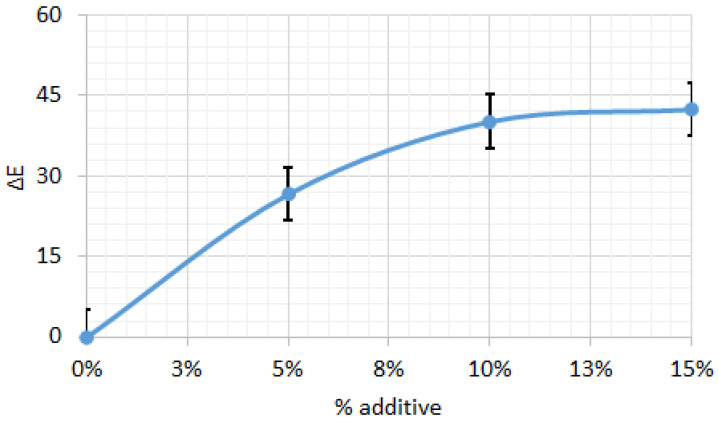
Color difference for biscuit dough between control sample and those with different percentages of additive. All data have statistically significant difference at *p* < 0.05.

**Figure 5 foods-14-03924-f005:**
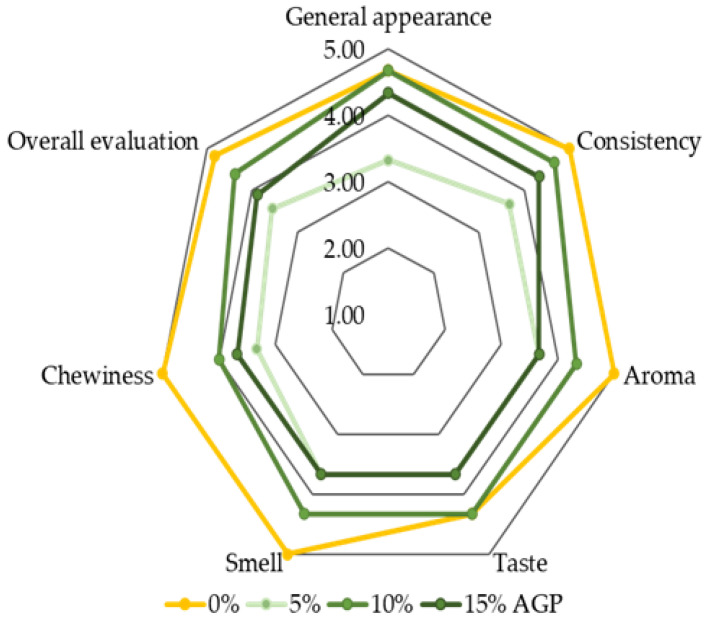
A 5-point Likert scale: 1—completely does not correspond to the indicator; 5—completely corresponds to the indicator; AGP—amaranth green powder.

**Figure 6 foods-14-03924-f006:**
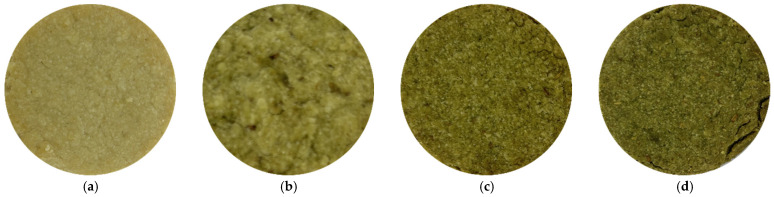
Color images of biscuits with pigweed powder. (**a**) 0% additive; (**b**) 5% additive; (**c**) 10% additive; (**d**) 15% additive.

**Figure 7 foods-14-03924-f007:**
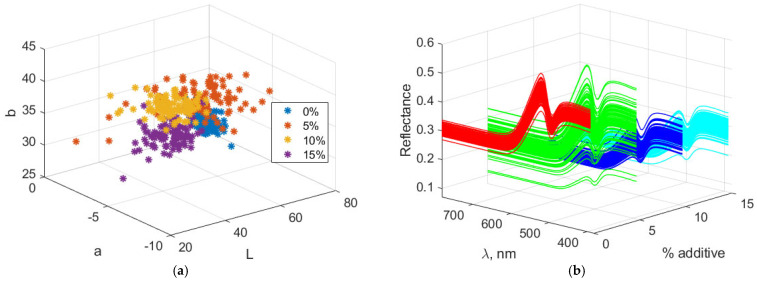
Color and spectral characteristics of biscuits with amaranth powder. (**a**) Lab color characteristics; (**b**) Spectral characteristics.

**Figure 8 foods-14-03924-f008:**
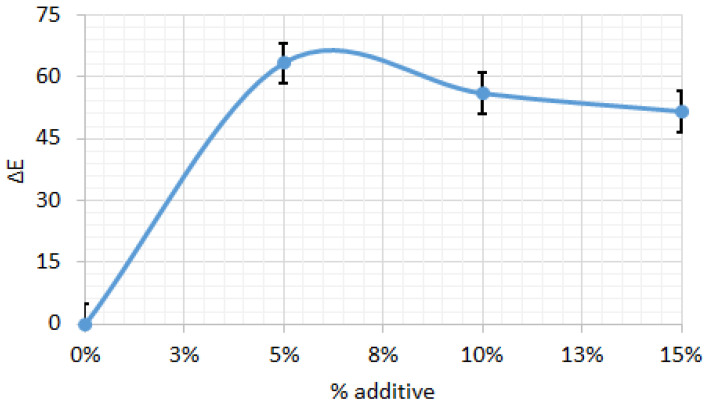
Color difference for biscuits between control sample and those with different percentage of amaranth powder. All data have statistically significant difference at *p* < 0.05.

**Figure 9 foods-14-03924-f009:**
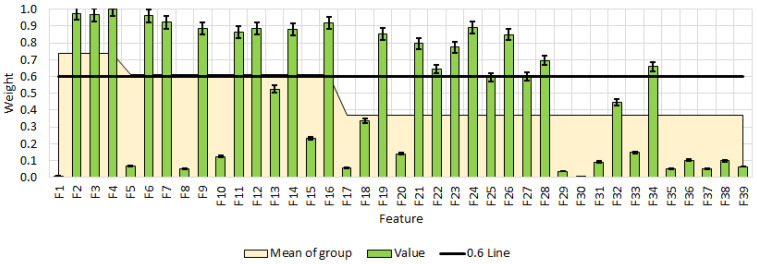
Feature selection by RReliefF method. All data have statistically significant difference at *p* < 0.05. F1—pH; F2—TDS; F3—EC; F4—ORP; F5—pH; F6—TDS; F7—EC; F8—ORP; F9—C1; F10—C2; F11—C3; F12—C4; F13—S1; F14—S2; F15—S3; F16—dE; F17—pH; F18—TDS; F19—EC; F20—ORP; F21—C1; F22—C2; F23—C3; F24—C4; F25—S1; F26—S2; F27—S3; F28—dE; F29—D; F30—h; F31—SF; F32—TL; F33—GA; F34—Color; F35—Aroma; F36—Taste; F37—Smell; F38—Chewiness; F39—OA.

**Figure 10 foods-14-03924-f010:**
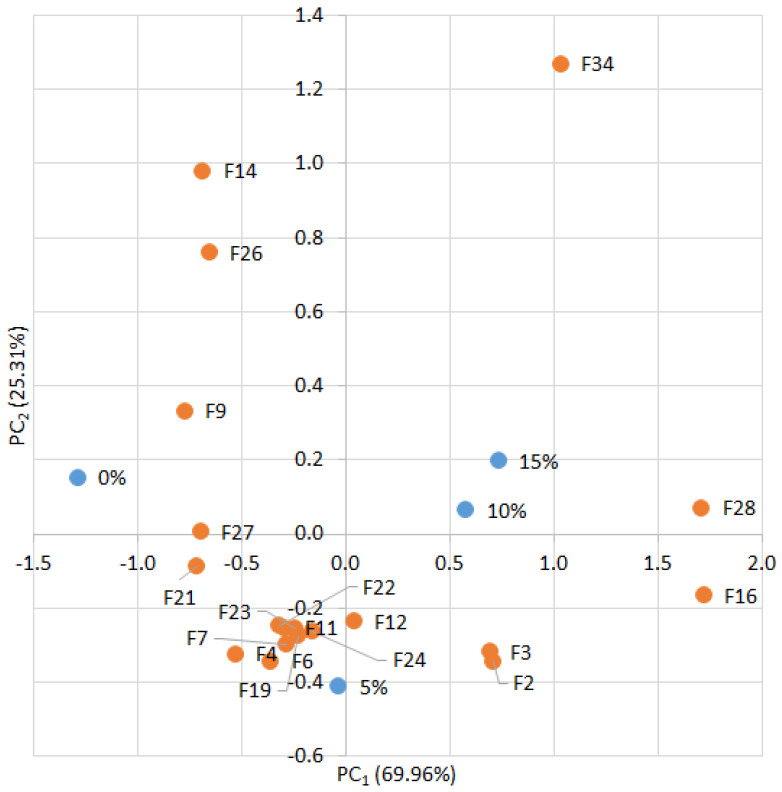
PCA of connections % additive/feature. F1—pH; F2—TDS; F3—EC; F4—ORP; F5—pH; F6—TDS; F7—EC; F8—ORP; F9—C1; F10—C2; F11—C3; F12—C4; F13—S1; F14—S2; F15—S3; F16—dE; F17—pH; F18—TDS; F19—EC; F20—ORP; F21—C1; F22—C2; F23—C3; F24—C4; F25—S1; F26—S2; F27—S3; F28—dE; F29—D; F30—h; F31—SF; F32—TL; F33—GA; F34—Color; F35—Aroma; F36—Taste; F37—Smell; F38—Chewiness; F39—OA.

**Figure 11 foods-14-03924-f011:**
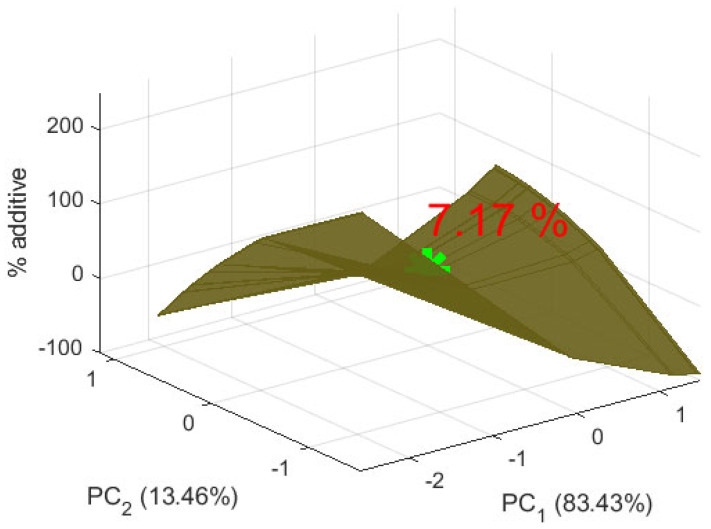
Appropriate amount of additive in biscuits.

**Table 1 foods-14-03924-t001:** Ingredients of the salty butter biscuits with amaranth green powder.

	Biscuit Type	0%	5%	10%	15%
Ingredient Amount					
Wheat flour, g		100	95	90	85
Amaranth green powder, g		0	5	10	15
Cow butter, g		50	50	50	50
Salt, g		1	1	1	1
Egg yolk, g		10	10	10	10

**Table 2 foods-14-03924-t002:** Features and their meanings.

Feature	Meaning	Feature	Meaning	Feature	Meaning
Flour		F13	S1	F26	S2
F1	pH	F14	S2	F27	S3
F2	TDS	F15	S3	F28	dE
F3	EC	F16	dE	F29	D
F4	ORP	Biscuits		F30	h
Dough		F17	pH	F31	SF
F5	pH	F18	TDS	F32	TL
F6	TDS	F19	EC	F33	GA
F7	EC	F20	ORP	F34	Color
F8	ORP	F21	C1	F35	Consistency
F9	C1	F22	C2	F36	Aroma
F10	C2	F23	C3	F37	Taste
F11	C3	F24	C4	F38	Smell
F12	C4	F25	S1	F39	Chewiness

pH—active acidity; TDS—totally dissolved solids; EC—electrical conductivity; ORP—oxidation reduction potential; Cx—color indices (x = 1 … 4); Sx—spectral indices (x = 1 … 3); dE—color difference; h—height; SF—spread factor; TL—thermal loses; GA—general appearance.

**Table 3 foods-14-03924-t003:** Feature/%AWF combinations table.

	Feature	F1	F2	…	Fm
% Additive					
0%		0%/F1	0%/F2	…	0%/Fm
5%		5%/F1	5%/F2	…	5%/Fm
…		…	…	…	…
15%		15%/F1	15%/F2	…	15%/Fm

**Table 4 foods-14-03924-t004:** Main characteristics of the green powder from the used amaranth species.

	Species	*A. albus*	*A. blitum*	*A. deflexus*	*A. hybridus*	*A. retroflexus*
Characteristic						
Radical scavenging capacity		3.3 ± 0.1	3.3 ± 0.1	3.6 ± 0.1	4.5 ± 0.2	3.8 ± 0.1
Total phenolic content		0.5 ± 0.1	0.9 ± 0.1	0.2 ± 0.1	1.5 ± 0.1	1.4 ± 0.1
Total flavonoid content		23.9 ± 0.9	18.2 ± 0.8	23.9 ± 1.0	22.3 ± 0.9	29.5 ± 1.1
Total condensed tannins		18.0 ± 0.4	13.8 ± 0.4	18.8 ± 0.5	13.2 ± 0.4	12.1 ± 0.3
Ca, mg/kg		27,885.05 ± 4182.76	27,489.84 ± 4123.48	19,286.70 ± 2893.01	26,453.34 ± 3968.00	24,520.49 ± 3678.07
P, %		0.43 ± 0.065	0.68 ± 0.102	0.44 ± 0.066	0.54 ± 0.081	0.50 ± 0.075
K, mg/kg		40,963.83 ± 6144.57	44,111.52 ± 6616.73	41,506.23 ± 6225.93	41,920.34 ± 6288.05	41,165.63 ± 6174.84
Mg, mg/kg		7461.65 ± 1119.25	9340.37 ± 1401.06	8117.06 ± 1217.56	6902.10 ± 1035.32	6768.25 ± 1015.24
Mn, mg/kg		49.14 ± 7.37	69.12 ± 10.37	68.93 ± 10.34	47.68 ± 7.15	32.24 ± 4.84
Zn, mg/kg		57.20 ± 8.58	72.78 ± 10.92	50.55 ± 7.58	34.85 ± 5.23	36.96 ± 5.54
Cu, mg/kg		11.48 ± 1.72	21.29 ± 3.19	23.69 ± 3.55	10.01 ± 1.50	10.48 ± 1.57
Fe, mg/kg		287.23 ± 43.08	401.53 ± 60.23	636.60 ± 95.49	486.76 ± 73.01	174.75 ± 26.21
Moisture, %		4.37 ± 0.66	3.62 ± 0.54	4.11 ± 0.62	4.44 ± 0.67	4.18 ± 0.63
DW, %		95.63 ± 14.34	96.38 ± 14.46	95.89 ± 14.38	95.56 ± 14.33	95.82 ± 14.37
Crude protein, %		21.88 ± 3.28	24.94 ± 3.74	18.50 ± 2.78	15.75 ± 2.36	22.00 ± 3.30
Crude fats, %		0.81 ± 0.12	0.93 ± 0.14	1.43 ± 0.21	0.55 ± 0.08	0.62 ± 0.09
Fibers, %		9.80 ± 1.47	22.79 ± 3.42	22.45 ± 3.37	15.58 ± 2.34	10.21 ± 1.53
Ash, %		18.78 ± 2.82	24.13 ± 3.62	21.37 ± 3.21	19.86 ± 2.98	21.97 ± 3.30
Nitrogen-free extract, %		44.36 ± 6.65	23.59 ± 3.54	32.14 ± 4.82	43.82 ± 6.57	41.02 ± 6.15

**Table 5 foods-14-03924-t005:** Main physico-chemical characteristics of the amaranth green powder and flour mixtures.

	AGP, %	0%	5%	10%	15%	100%
Parameter						
pH		6.55 ± 0.06 ^ab^	6.46 ± 0.03 ^b^	6.37 ± 0.04 ^b^	6.28 ± 0.05 ^b^	6.67 ± 0.14 ^a^
TDS, ppm		627 ± 97 ^d^	1175 ± 104 ^d^	1920 ± 0.05 ^c^	2523 ± 243 ^b^	7621.67 ± 1303.74 ^a^
EC, µS/cm		1075 ± 108 ^d^	2016 ± 305 ^c^	3011 ± 0.06 ^c^	3759 ± 421 ^b^	8402.6 ± 1152.19 ^a^
ORP, mV		65 ± 19 ^ab^	67 ± 12 ^ab^	73 ± 0.07 ^a^	78 ± 18 ^a^	55.83 ± 9.89 ^b^

AGP—amaranth green powder. Values ± SD are statistically different at *p* ≤ 0.05 (ANOVA LSD post hoc test); Means in a row with different letters (a, b, c, and d) are significantly different (*p* ≤ 0.05, ANOVA LSD).

**Table 6 foods-14-03924-t006:** Main physico-chemical characteristics of biscuit dough. All data have statistically significant difference at *p* < 0.05.

	AGP, %	0%	5%	10%	15%
Parameter					
pH		6.24 ± 0.06 ^b^	6.35 ± 0.04 ^a^	6.34 ± 0.06 ^a^	6.34 ± 0.09 ^a^
TDS, ppm		1834.5 ± 3.5 ^d^	1892 ± 25 ^cd^	2277 ± 124 ^b^	2439 ± 32 ^a^
EC, µS/cm		2962.5 ± 59.5 ^d^	3152 ± 40 ^c^	3742 ± 58 ^b^	3932.5 ± 32.5 ^a^
ORP, mV		65.5 ± 3.5 ^a^	64 ± 2 ^a^	64.5 ± 1.5 ^a^	66.5 ± 0.5 ^a^

Means in a row with different letters (a, b, c, and d) are significantly different (*p* ≤ 0.05, ANOVA LSD).

**Table 7 foods-14-03924-t007:** Values of color indices of biscuits dough with additive. All data have statistically significant difference at *p* < 0.05.

	% Additive	0%	5%	10%	15%
Color Index					
C1		106.68 ± 0.55 ^a^	80.75 ± 0.65 ^b^	70.98 ± 0.77 ^c^	68.56 ± 0.81 ^c^
C2		120.92 ± 1.88 ^b^	118.23 ± 0.79 ^c^	124.05 ± 0.15 ^a^	121.68 ± 0.79 ^ab^
C3		73.14 ± 1.69 ^c^	81.10 ± 0.88 ^b^	91.06 ± 0.23 ^a^	91.00 ± 0.46 ^a^
C4		114.12 ± 2.71 ^d^	144.67 ± 1.81 ^c^	167.79 ± 0.84 ^b^	169.30 ± 1.09 ^a^

Means in a row with different letters (a, b, c, and d) are significantly different (*p* ≤ 0.05, ANOVA LSD).

**Table 8 foods-14-03924-t008:** Values of spectral indices of biscuits dough with additive. All data have statistically significant difference at *p* < 0.05.

	% Additive	0%	5%	10%	15%
Spectral Index					
S1		0.62 ± 0.02 ^a^	0.60 ± 0.02 ^ab^	0.61 ± 0.02 ^ab^	0.60 ± 0.30 ^b^
S2		0.87 ± 0.03 ^a^	0.50 ± 0.02 ^b^	0.25 ± 0.01 ^c^	0.24 ± 0.01 ^c^
S3		0.24 ± 0.01 ^a^	0.25 ± 0.01 ^a^	0.24 ± 0.01 ^a^	0.25 ± 0.01 ^a^

Means in a row with different letters (a, b and c) are significantly different (*p* ≤ 0.05, ANOVA LSD).

**Table 9 foods-14-03924-t009:** Main physico-chemical characteristics of biscuits with Amaranth green powder. All data have statistically significant difference at *p* < 0.05.

	% Additive	0%	5%	10%	15%
Characteristic					
pH		6.34 ± 0.15 ^a^	6.24 ± 0.06 ^b^	6.24 ± 0.05 ^b^	6.19 ± 0 ^b^
TDS, ppm		1794 ± 13 ^b^	1948 ± 161 ^b^	2314 ± 249 ^a^	2266 ± 168 ^a^
EC, µS/cm		3189 ± 66 ^b^	3551 ± 19 ^b^	4232 ± 15 ^a^	4128 ± 33 ^a^
ORP, mV		83.5 ± 5.5 ^a^	83.0 ± 5.0 ^a^	83 ± 4 ^a^	83 ± 3.0 ^a^

Means in a row with different letters (a and b) are significantly different (*p* ≤ 0.05, ANOVA LSD).

**Table 10 foods-14-03924-t010:** The geometric properties and thermal losses of butter biscuits with amaranth powder. All data have statistically significant difference at *p* < 0.05.

	% Additive	0%	5%	10%	15%
Characteristic					
D, mm		47.87 ± 1.56 ^a^	48.15 ± 1.21 ^a^	47.35 ± 1.59 ^a^	47.85 ± 1.46 ^a^
h, mm		10.29 ± 0.75 ^b^	11.12 ± 0.79 ^ab^	11.39 ± 0.65 ^a^	10.18 ± 0.25 ^b^
SF		4.65 ± 2.08 ^a^	4.33 ± 1.53 ^a^	4.16 ± 2.46 ^a^	4.7 ± 5.79 ^a^
TL, %		15	17	18	17

Means in a row with different letters (a and b) are significantly different (*p* ≤ 0.05, ANOVA LSD).

**Table 11 foods-14-03924-t011:** Average sensory scores of salty butter biscuits with different amounts of amaranth green powder (*n* = 9). All data have statistically significant difference at *p* < 0.05.

	% Additive	0%	5%	10%	15%
Characteristic					
General appearance		4.67 ± 0.58 ^ab^	3.33 ± 2.08 ^c^	4.67 ± 0.58 ^ab^	4.33 ± 1.15 ^b^
Consistency		5 ± 0 ^a^	3.67 ± 2.31 ^bc^	4.67 ± 0.58 ^ab^	4.33 ± 1.15 ^b^
Aroma		5 ± 0 ^a^	3.67 ± 2.31 ^bc^	4.33 ± 0.58 ^b^	3.67 ± 0.58 ^c^
Taste		4.33 ± 0.58 ^ab^	3.67 ± 2.31 ^c^	4.33 ± 0.58 ^ab^	3.67 ± 0.58 ^c^
Smell		5 ± 0 ^a^	3.67 ± 2.31 ^bc^	4.33 ± 0.58 ^ab^	3.67 ± 0.58 ^c^
Chewiness		5 ± 0 ^a^	3.33 ± 2.08 ^c^	4 ± 0 ^b^	3.67 ± 0.58 ^c^
Overall evaluation		4.83 ± 0.19 ^a^	3.56 ± 2.23 ^bc^	4.39 ± 0.48 ^ab^	3.89 ± 0.77 ^b^

Means in a row with different letters (a, b and c) are significantly different (*p* ≤ 0.05, ANOVA LSD).

**Table 12 foods-14-03924-t012:** Values of color indices of biscuits with amaranth powder. All data have statistically significant difference at *p* < 0.05.

	% Additive	0%	5%	10%	15%
Color Index					
C1		102.36 ± 0.44 ^b^	102.74 ± 0.79 ^b^	97.94 ± 0.48 ^c^	92.49 ± 0.58 ^d^
C2		108.22 ± 2.04 ^d^	125.85 ± 1.62 ^c^	135.92 ± 0.72 ^b^	130.66 ± 0.65 ^bc^
C3		72.15 ± 1.75 ^d^	83.71 ± 2.05 ^c^	93.8 ± 0.75 ^b^	92.13 ± 0.47 ^b^
C4		115.33 ± 2.98 ^d^	134.36 ± 3.86 ^c^	154.38 ± 1.16 ^b^	155.23 ± 0.48 ^a^

Means in a row with different letters (a, b, c, and d) are significantly different (*p* ≤ 0.05, ANOVA LSD).

**Table 13 foods-14-03924-t013:** Values of spectral indices of biscuits with amaranth powder. All data have statistically significant difference at *p* < 0.05.

	% Additive	0%	5%	10%	15%
Spectral Index					
S1		0.62 ± 0.02 ^b^	0.64 ± 0.02 ^ab^	0.66 ± 0.02 ^a^	0.65 ± 0.02 ^a^
S2		0.95 ± 0.04 ^a^	0.67 ± 0.04 ^b^	0.41 ± 0.01 ^c^	0.42 ± 0.01 ^c^
S3		0.24 ± 0.01 ^a^	0.22 ± 0.01 ^a^	0.20 ± 0.01 ^a^	0.21 ± 0.01 ^a^

Means in a row with different letters (a, b and c) are significantly different (*p* ≤ 0.05, ANOVA LSD).

**Table 14 foods-14-03924-t014:** Normed values of % additive/feature combinations.

	% Additive	0%	5%	10%	15%
Feature					
F2		0.00	0.22	0.51	0.75
F3		0.00	0.25	0.52	0.71
F4		0.00	0.03	0.10	0.17
F6		0.02	0.02	0.18	0.25
F7		0.36	0.06	0.21	0.26
F9		0.02	0.12	0.03	0.00
F11		0.02	0.11	0.22	0.22
F12		0.71	0.20	0.33	0.34
F14		0.00	0.30	0.01	0.01
F16		0.02	0.62	0.93	0.98
F19		0.10	0.10	0.26	0.24
F21		0.01	0.10	0.06	0.00
F22		0.01	0.14	0.21	0.17
F23		0.01	0.13	0.24	0.22
F24		0.57	0.14	0.26	0.27
F26		0.17	0.29	0.02	0.03
F27		0.00	0.11	0.03	0.06
F28		0.80	0.98	0.87	0.80
F34		0.00	0.53	0.73	0.67

F1—pH; F2—TDS; F3—EC; F4—ORP; F5—pH; F6—TDS; F7—EC; F8—ORP; F9—C1; F10—C2; F11—C3; F12—C4; F13—S1; F14—S2; F15—S3; F16—dE; F17—pH; F18—TDS; F19—EC; F20—ORP; F21—C1; F22—C2; F23—C3; F24—C4; F25—S1; F26—S2; F27—S3; F28—dE; F29—D; F30—h; F31—SF; F32—TL; F33—GA; F34—Color; F35—Aroma; F36—Taste; F37—Smell; F38—Chewiness; F39—OA.

**Table 15 foods-14-03924-t015:** Assessment of regression model *%A* = *f*(*PC*_1_, *PC*_2_).

N = 122	R = 0.99; R^2^ = 0.99; Adjusted R^2^ = 0.99; F(5, 6) = 611.20; *p* < 0.00; SE = 0.35					
	b	SE of b	b	SE of b	t(6)	*p*-Value
Intercept	-	-	−0.06	0.64	−5.09	0.01
*PC* _1_	1.25	0.06	8.77	0.39	22.26	0.00
*PC* _2_	−1.42	0.18	−28.15	3.52	−8.01	0.00
*PC* _1_ ^2^	1.74	0.16	15.71	1.44	10.93	0.00
*PC* _2_ ^2^	−0.83	0.11	−29.92	3.95	−7.59	0.00
*PC* _1_ *PC* _2_	1.17	0.13	49.10	5.48	8.96	0.00

## Data Availability

The original contributions presented in this study are included in the article. Further inquiries can be directed to the corresponding author.

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
