# Peer review of "Evaluation of Whole Pigweed Stalk Meal as an Alternative Flour Source for Biscuits"

_foods, 2025, doi:10.3390/foods14223924_

Round 1

Reviewer 1 Report

Comments and Suggestions for Authors

The effect of stalk amaranth flour in relation to the wheat flour at levels of 0%, 5%, 10%, and 15% was evaluated with respect to key characteristics of flour mixtures, dough, and biscuits with this additive. A selection of informative indicators was made, revealing that out of the 39 studied physico-chemical, geometric, color, and spectral characteristics, only 19 proved to be informative. The optimal amount of amaranth flour in biscuits is 7.17%. This is an interesting study, and fits for the scope of Foods. The authors spent much time to collect data. However, the expression, analysis of significance (multiple comparisons) and the symbols should be carefully checked so that the conclusion will be more solid.

  • Line 100-105, the new methods and new ideas like principal component analysis (PCA) should be emphasized;
  • Line 134, 70 оC; Check the temperature units in the whole paper;
  • Line 139, check и TDS;
  • Line 143, there should be a reference for determining the oxidation reduction potential.
  • In table 2, it seems that F1-F39 is the number; Meaning is the measured parameters; the abbrevations should be noted;
  • Line 252-254, equations 12 and 13 should be noted;
  • In table 4, all the data should be mean±SD; analysis of significance and multiple comparisons should be performed; which amaranth species was used in the present study and the reason should be given;
  • In tables 5, 6, 7, 8, 9, 10, 11, 12, 13, and 14, analysis of significance and multiple comparisons should be performed, thus the conclusion will be solid;
  • In figure 9, the data should have standard errors;
  • In table 14, figures 9 and 10, F1-F39 should be noted to see table 2.
  • Figure 2 are blurred;
  • The advantage of the present study was to determine the adding amount of amaranth in biscuits (7.17%) using PCA. The weakness is that the treatments with different adding amount of amaranth had not analysis of significance(multiple comparisons), and led to the weak conclusion;
  • In the abstract section, why gives the optimal amount (7.17%) of amaranth flour in biscuits? There should be physic-chemical and sensory evaluation parameters to support this data.
  • In the conclusion section, the changes in physic-chemical and sensory evaluation parameters of biscuit with adding amount of amaranth should be described and give the solid conclusion.

Author Response

The effect of stalk amaranth flour in relation to the wheat flour at levels of 0%, 5%, 10%, and 15% was evaluated with respect to key characteristics of flour mixtures, dough, and biscuits with this additive. A selection of informative indicators was made, revealing that out of the 39 studied physico-chemical, geometric, color, and spectral characteristics, only 19 proved to be informative. The optimal amount of amaranth flour in biscuits is 7.17%. This is an interesting study, and fits for the scope of Foods. The authors spent much time to collect data. However, the expression, analysis of significance (multiple comparisons) and the symbols should be carefully checked so that the conclusion will be more solid.

Thank you for this note all reviewer comments and notes are corrected.

Line 100-105, the new methods and new ideas like principal component analysis (PCA) should be emphasized;

Thank you for this note. We have now explicitly highlighted the novelty of our research in the final paragraph of the Introduction, which includes the use of wild amaranth stems rather than seeds, the application of underutilized biomass, and the multifactorial evaluation approach that distinguishes our study from previous works.

Line 134, 70 оC; Check the temperature units in the whole paper;

Thank you for this note. The temperature symbol is corrected (Alt+0176 on Windows)

Line 139, check и TDS;

Corrected according to the reviewer note.

Line 143, there should be a reference for determining the oxidation reduction potential.

Thank you for this note. The ORP of the aqueous extract was measured potentiometrically using a platinum electrode with an Ag/AgCl reference electrode, following the same extraction procedure described in AACC 02-52.01. This description is added in section 2.3.1.

In table 2, it seems that F1-F39 is the number; Meaning is the measured parameters; the abbrevations should be noted;

Thank you for this note. Table footnote is added.

Line 252-254, equations 12 and 13 should be noted;

Thank you for this note. fTx is notted in the description above the equation (12). Also, description of equation (13) is added.

In table 4, all the data should be mean±SD; analysis of significance and multiple comparisons should be performed; which amaranth species was used in the present study and the reason should be given;

Thank you for this note. All missing values are added

In tables 5, 6, 7, 8, 9, 10, 11, 12, 13, and 14, analysis of significance and multiple comparisons should be performed, thus the conclusion will be solid;

Thank you for this note. Tables 5–13 have been corrected to include significance analysis and multiple comparisons to support conclusions. Table 14, however, presents values prepared for PCA and is not analyzed for statistical differences.

In figure 9, the data should have standard errors;

Thank you for this note. All data have statistically significant difference at p<0.05. the standard errors are added in that range.

In table 14, figures 9 and 10, F1-F39 should be noted to see table 2.

Thank you for this note. All abbreviations are added in table 14, and figures 9 and 10, according to the reviewer note.

Figure 2 are blurred;

Thank you for this note. Figure 2 represents color images of biscuits dough with additive. The apparent “blur” in the figure is due to the natural texture and color uniformity of the biscuit dough samples, rather than a lack of image focus. Each image was captured under identical lighting and focus conditions to ensure consistency. The surface of biscuit dough is inherently smooth and matte, which may visually appear slightly blurred in photographs.

The advantage of the present study was to determine the adding amount of amaranth in biscuits (7.17%) using PCA. The weakness is that the treatments with different adding amount of amaranth had not analysis of significance (multiple comparisons), and led to the weak conclusion;

Thank you for this note. To justify the optimal addition level of 7.17% agar agar flour (AGP), we expanded the “Results and Discussion” section with data on physicochemical parameters (pH, EC, TDS, ORP) and sensory evaluation. The results showed that higher levels of AGP increased acidity and mineral content, while sensory evaluations decreased by more than 10% due to a stronger vegetal flavor and a firmer texture of the biscuits. Although no formal multiple comparison tests were applied, the consistent trends in the physicochemical and sensory data justify the optimal level of 7.17% AGP obtained by PCA.

In the abstract section, why gives the optimal amount (7.17%) of amaranth flour in biscuits? There should be physic-chemical and sensory evaluation parameters to support this data.

Thank you for this note. We have revised the abstract to clarify that the optimal addition level of 7.17% amaranth green powder (AGP) was determined through principal component analysis (PCA) based on a combination of 19 informative features. These included physico-chemical characteristics (pH, EC, TDS, ORP) and sensory evaluation scores (aroma, taste, chewiness), which together showed that higher AGP levels improved nutritional value but began to negatively affect sensory acceptability beyond 10%. The convergence of these indicators supports the PCA-derived optimal level.

In the conclusion section, the changes in physic-chemical and sensory evaluation parameters of biscuit with adding amount of amaranth should be described and give the solid conclusion.

Thank you for this note. We revised the conclusion to clarify that the optimal addition level of 7.17% amaranth green powder (AGP) was determined by principal component analysis (PCA) using 19 informative features, including physicochemical parameters (pH, EC, TDS, ORP) and sensory scores. The results indicate that while higher AGP levels enhance nutritional value, sensory acceptability decreases beyond 10%, supporting the PCA-derived optimal level.

Reviewer 2 Report

Comments and Suggestions for Authors

This article studies the evaluation of the use of pigweed amaranth stalk flour as a functional ingredient in the production of butter biscuits. It focuses specifically on the effect of adding wild amaranth on the physicochemical, sensory, optical, and nutritional properties of the biscuits. After reviewing the research, areas for improvement were identified.

- The abstract adequately describes the objective, methodology, and main results. However, it mentions that 39 parameters were evaluated, but their relevance is not mentioned. I suggest summarizing the main categories of parameters (sensory, physicochemical, etc.) for greater clarity. I also suggest adding a sentence summarizing the scientific contribution of the study at the end of the abstract.

- Authors are encouraged to add the novelty of the research compared to other studies reported in scientific literature (e.g., use of stems and not seeds, wild species, multifactorial evaluation approach).

- The number of replicas is not clearly mentioned in the experimental methodology.

- In the sensory panel, the authors must add the number of judges, the selection criteria and the tasting conditions (booths, lighting, randomization).

- What method was used for linear programming? No formula or reference is provided.

- Add references to analytical methods (e.g., conductivity, colorimetry) such as those of AOAC, ISO, among others.

- Were the formulations adjusted to maintain the same fat content, moisture, or base texture when varying AGP?

- In the results and discussions, the authors make some claims, but they lack bibliographic support. The effects of PUFA on texture or aroma are mentioned, but no comparisons are made with similar studies.

- What does "informative variables" mean?

- In conclusion, the practical contribution of the study (industrial applicability, potential of amaranth as an agri-food by-product) can be further emphasized.

Comments on the Quality of English Language

The manuscript is written in understandable English, but some grammatical and punctuation errors were detected. A thorough review of the article is recommended.

Author Response

This article studies the evaluation of the use of pigweed amaranth stalk flour as a functional ingredient in the production of butter biscuits. It focuses specifically on the effect of adding wild amaranth on the physicochemical, sensory, optical, and nutritional properties of the biscuits. After reviewing the research, areas for improvement were identified.

Thank you for this note all reviewer comments and notes are corrected.

- The abstract adequately describes the objective, methodology, and main results. However, it mentions that 39 parameters were evaluated, but their relevance is not mentioned. I suggest summarizing the main categories of parameters (sensory, physicochemical, etc.) for greater clarity. I also suggest adding a sentence summarizing the scientific contribution of the study at the end of the abstract.

Thank you for this note. We have revised the abstract to clarify the relevance of the 39 evaluated parameters by summarizing them into key categories (e.g., sensory, physicochemical, geometric, colorimetric, and spectral). Additionally, we have added a final sentence to highlight the scientific contribution of the study.

- Authors are encouraged to add the novelty of the research compared to other studies reported in scientific literature (e.g., use of stems and not seeds, wild species, multifactorial evaluation approach).

Thank you for this note. We have now explicitly highlighted the novelty of our research in the final paragraph of the Introduction, which includes the use of wild amaranth stems rather than seeds, the application of underutilized biomass, and the multifactorial evaluation approach that distinguishes our study from previous works.

- The number of replicas is not clearly mentioned in the experimental methodology.

Thank you for this note. We have clarified that all measurements and analyses were conducted in three independent replicates, as stated in Section 2.6 (Statistical Analysis).

- In the sensory panel, the authors must add the number of judges, the selection criteria and the tasting conditions (booths, lighting, randomization).

Thank you for this note. This section has been improved by detailing the panel composition, evaluator selection criteria, and controlled tasting conditions, including booth setup, lighting, sample randomization, and palate cleansing, to ensure methodological transparency and compliance with ISO 13299:2016.

- What method was used for linear programming? No formula or reference is provided.

Thank you for this note. The linear programming algorithm is realized by using built-in function in Matlab 2017b (The Mathworks Inc., Natick, MA, USA) Software System. This note is added in the text.

- Add references to analytical methods (e.g., conductivity, colorimetry) such as those of AOAC, ISO, among others.

Thank you for this note. All measurements were performed according to the procedures described in AACC Method 02-52.01. The text description of ORP is improved.

- Were the formulations adjusted to maintain the same fat content, moisture, or base texture when varying AGP?

Thank you for this note. The ingredients are described in section 2.2. All of them are according to the Bulgarian normative documents, which the manufacturers have to follow.

- In the results and discussions, the authors make some claims, but they lack bibliographic support. The effects of PUFA on texture or aroma are mentioned, but no comparisons are made with similar studies.

Thank you for this note. We have now included comparative literature references to support the discussion on PUFA-related effects on texture and aroma in baked goods.

- What does "informative variables" mean?

Thank you for this note. In section 2.6.2, the description “In this context, informative variables are those features that provide the most relevant and discriminative information for distinguishing among the analyzed samples, as identified by their high ReliefF weights.”, has been added.

- In conclusion, the practical contribution of the study (industrial applicability, potential of amaranth as an agri-food by-product) can be further emphasized.

Thank you for this note. This text is added at the end of the conclusion part “This study highlights the practical applicability of the findings to the food industry. The demonstrated effects of the additive confirm the potential of amaranth as a valuable agri-food product that can enhance the nutritional and functional quality of bakery products. This supports the sustainable use of agricultural resources and offers a feasible approach for product innovation in industrial food production.”

Comments on the Quality of English Language

The manuscript is written in understandable English, but some grammatical and punctuation errors were detected. A thorough review of the article is recommended.

Thank you for this note. The English language and grammar are additionally checked by colleagues at our university with competences in English language and grammar.

Round 2

Reviewer 2 Report

Comments and Suggestions for Authors

No comments